

# Developing the Leuven Embedded Figures Test (L-EFT): testing the stimulus features that influence embedding

Lee de-Wit[1,2,3,*], Hanne Huygelier[3,*], Ruth Van der Hallen[3], Rebecca Chamberlain[3] and Johan Wagemans[3]

[1] Institute of Continuing Education, University of Cambridge, Cambridge, United Kingdom
[2] Department of Psychology and Language Sciences, University College London, London, United Kingdom
[3] Laboratory of Experimental Psychology, Katholieke Universiteit Leuven, Leuven, Belgium
[*] These authors contributed equally to this work.

## ABSTRACT

**Background.** The Embedded Figures Test (EFT, developed by Witkin and colleagues (*1971*)) has been used extensively in research on individual differences, particularly in the study of autism spectrum disorder. The EFT was originally conceptualized as a measure of field (in)dependence, but in recent years performance on the EFT has been interpreted as a measure of local versus global perceptual style. Although many have used the EFT to measure perceptual style, relatively few have focused on understanding the stimulus features that cause a shape to become embedded. The primary aim of this work was to investigate the relation between the strength of embedding and perceptual grouping on a group level.

**Method.** New embedded figure stimuli (both targets and contexts) were developed in which stimulus features that may influence perceptual grouping were explicitly manipulated. The symmetry, closure and complexity of the target shape were manipulated as well as its good continuation by varying the number of lines from the target that continued into the context. We evaluated the effect of these four stimulus features on target detection in a new embedded figures task (Leuven Embedded Figures Test, L-EFT) in a group of undergraduate psychology students. The results were then replicated in a second experiment using a slightly different version of the task.

**Results.** Stimulus features that influence perceptual grouping, especially good continuation and symmetry, clearly affected performance (lower accuracy, slower response times) on the L-EFT. Closure did not yield results in line with our predictions.

**Discussion.** These results show that some stimulus features, which are known to affect perceptual grouping, also influence how effectively a stimulus becomes embedded in different contexts. Whether these results imply that the EFT measures individual differences in perceptual grouping ability must be further investigated.

Corresponding author
Lee de-Wit,
Lee.de-Wit@ice.cam.ac.uk

## INTRODUCTION

Visual processing involves more than mere linear summations of visual input. This particular insight, that the whole is different than the sum of its parts, is one of the key principles of Gestalt psychology (*Wagemans et al., 2012*). It has become increasingly clear, however, that whilst this principle constitutes a universal feature of visual information processing, there are also pronounced inter-individual differences (*De-Wit & Wagemans, 2015*).

These individual differences were first conceptualized by Witkin and colleagues in terms of *field dependence* and *field independence* (*Witkin et al., 1954*). Field dependence refers to a cognitive style in which the perception of a local element is influenced by the surrounding context, while this is not (or much less) the case in field independence (*Goodenough & Witkin, 1977*; *Witkin et al., 1954*; *Witkin et al., 1975*). The concept of perceptual style was in part motivated by the observation of individual differences in the *rod-and-frame test*, in which some observers' judgements of the orientation of a rod proved to be dependent on the frame surrounding it, whilst some observers could make judgements of the rod's orientation independent from the frame (*Goodenough & Witkin, 1977*; *Witkin, 1950*). The construct of field dependence was then further supported by the observation that observers who could judge the orientation of the rod independent of the frame were also better at the Gottschaldt embedded figures (*Adevai, Silverman & Gough, 1968*; *Witkin, 1950*).

Later on, Witkin and colleagues developed the *Embedded Figures Test* (EFT; *Witkin et al., 1971*) to measure field (in)dependence. In their EFT, participants are required to locate and trace the outline of a target (a simple closed shape) within an embedding context (a larger, more complex line pattern). The simple shape becomes difficult to detect by incorporating it as a part of the embedding context that then constitutes a perceived whole. Consequently, the complex line pattern dominates perception and the target shape becomes hidden or "embedded" within the context (*Goodenough & Witkin, 1977*).

Witkin's EFT has been used extensively in research on individual differences in subsequent years, particularly in the study of autism spectrum disorders and often to measure local versus global perceptual style (e.g., *Jolliffe & Baron-Cohen, 1997*; *Cribb et al., 2016*; *Panton, Badcock & Badcock, 2016*). Yet, relatively few studies have focused on actually understanding the stimulus features that drive *embedding*. Prior to Witkin's development of the EFT as a tool to measure individual differences, *Gottschaldt (1926)* had identified that the *familiarity* of a figure did not influence the extent to which a shape would become effectively embedded. Beyond the fact that familiarity had no influence on embedding strength, little is known about what really causes a 'part' to become embedded in a larger 'whole.'

Visual inspection of Witkin's embedded figures suggests that target shapes were embedded within the contexts on the basis of a number of different factors, amongst which numerous related to perceptual grouping. For instance, shaded areas were used to evoke grouping of particular regions in a way that could hinder target detection. For other figures, target detection was complicated by including parallel lines and/or line repetitions which create strong patterns that may dominate perception (and for some shapes may even create 3D percepts). Clearly, some aspects of *perceptual grouping* are involved in stimulus embedding, but perceptual grouping is a broad concept that could be underpinned by many

different levels of processing or perceptual effects (*De-Wit & Wagemans, 2015*; *Wagemans et al., 2012*). Furthermore, these potential factors used to embed the target shapes were not explicitly manipulated nor discussed, and therefore it remains unclear to what extent and in what way different perceptual grouping factors may influence perceived embedding.

There are a wide range of potential visual properties that could influence perceptual grouping, for example the original work of *Wertheimer* (*1923*) proposed factors including *proximity*, *similarity* and *good continuation*. Later research has confirmed that many of the principles first proposed by Wertheimer influence visual detectability. For example, *Prinzmetal & Banks (1977)* have shown that good continuation can affect visual detection, even with brief presentation of stimuli. Further work has also suggested additional grouping factors (*Wagemans et al., 2012*), for instance, *mirror symmetry* is a factor that has a small positive effect on figure-ground segmentation (*Machilsen, Pauwels & Wagemans, 2009*). Closure has also been proposed as an additional grouping factor (*Elder & Zucker, 1993*; *Kovács & Julesz, 1993*), but it is unclear whether closure has an influence beyond proximity and good continuation (*Tversky, Geisler & Perry, 2004*). So far, it is unclear which, if any, of these factors influence perceptual embedding.

## Current study

Although research regarding the EFT has traditionally focused on inter-individual differences, the primary aim of this work was to design a new embedded figures task (*Leuven Embedded Figures Test*, L-EFT), in which specific stimulus features were explicitly manipulated in order to investigate the role of perceptual grouping in embedding on a group level. With respect to the target, we manipulated a number of factors which, based on previous research, were expected to influence the detectability of the target. More specifically, we manipulated the symmetry and closure of the targets with the prediction that closed, symmetric shapes would be easier to detect than open, non-symmetric shapes. Additionally, we manipulated the complexity of the target shapes (simply defined as the number of lines that make up the target), with the expectation that simple shapes would me more easily detected than complex shapes.

With respect to the embedding context, we explored to what extent the number of lines of the target shape that continued into the context would determine the degree of embedding (Fig. 1E). We predicted that, in line with the general principle of good continuation, target shapes with more lines continued into the context would be more difficult to detect. This factor of good continuation was systematically manipulated across the different target types, such that each of the different target types was embedded in four different contexts with progressively increasing number of continued lines. This enabled us to test for the interaction between target and context embedding effects.

We decided to manipulate these factors within a new computerized version of the EFT, which would more easily enable us to reliably measure reaction time and more easily quantify accuracy. The format of this task was developed from the Leuven Perceptual Organization Screening Test (L-POST), using a matching to sample task (see Methods-Procedure) (*Torfs et al., 2013*). This format was adapted to form a new embedded figures test, which we will refer to as the Leuven Embedded Figures Test (L-EFT).

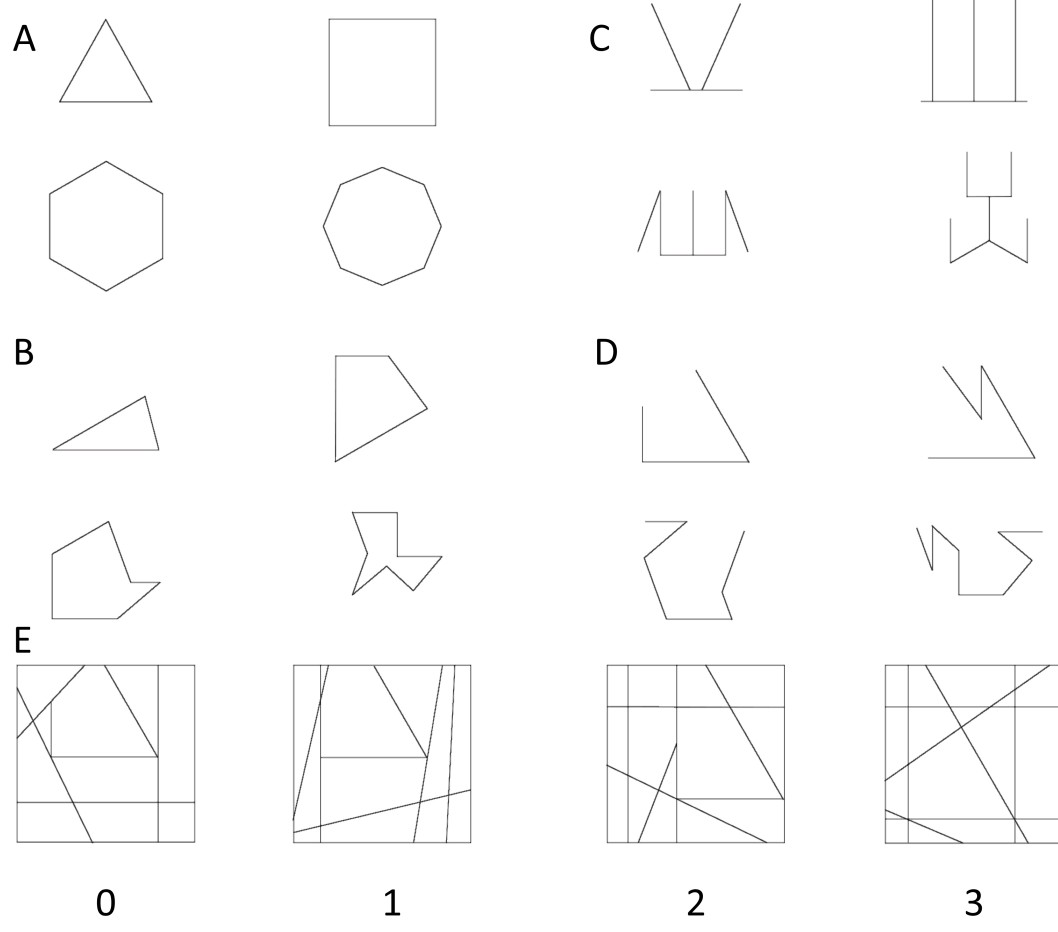

**Figure 1** L-EFT target shapes and embedding contexts with increasing levels of continued lines.

This study involved two separate waves of data collection (Experiment 1 and 2) in two large, non-overlapping samples of undergraduate students. Experiment 1 served to test which stimulus features would influence item difficulty on a group level, while Experiment 2 was conducted to replicate the findings of Experiment 1 with a new dataset and to verify whether the results would be robust to small changes in the task procedure.

# EXPERIMENT 1

## Materials & methods

### Participants

A sample of 255 undergraduate psychology students participated in this study for course credits. All participants were naïve to the purpose of the study. The median age of subjects was 19 years ($SD = 2.4$). The sample was primarily female (87%). All procedures performed in this study were in accordance with the ethical standards of the institutional ethical committee and approved by the ethical committee of the KU Leuven university (SMEC approval code: S58409) as well as in accordance with the 1964 Helsinki declaration

and its later amendments or comparable ethical standards. Written informed consent for each participant was obtained prior to testing.

### Stimuli

A total of 16 simple line drawings (targets) were presented which varied in the number of lines (3, 4, 6 and 8 lines), whether the target was symmetric around its vertical axis (Figs. 1A and 1C vs 1B and 1D) and whether the target formed an open (Figs. 1C and 1D) or closed shape (Figs. 1A and 1B). For each of these target shapes four complex line shapes (embedding contexts) were developed that varied in the number of target lines that were continued into the embedding contexts ranging from 0 lines to a maximum that is equal to the number of target lines (Fig. 1E). This combination of 16 shapes and 4 levels of good (line) continuation resulted in 64 trials. The embedding context always contained the target only at one location. For each trial, two additional contexts where constructed that did not contain the target shape (used as distractors, see Procedure). These distractors had the same number of lines as the embedding context that contained the target and no rotated or scaled versions of the target shapes. For some embedding contexts lines crossing the target shape were added (ranging from 2 to 5 lines), manipulated independently from closure and symmetry. No contexts had curved lines and care was taken to ensure that none of the targets or contexts would be perceived as 3D. The size of the contexts was constant. The L-EFT figures were created using an open source drawing program (Scribus). The complete stimulus set is made publicly available on Figshare (https://dx.doi.org/10.6084/m9.figshare.3807885, https://dx.doi.org/10.6084/m9.figshare.3807894).

### Procedure

The L-EFT consisted of 64 trials that were presented in a randomized order. For each trial, a matching-to-sample paradigm was used in which the participant was presented with the target (above) and three response options (below). Of these three response options, one contained the target, and two were distractor contexts (Fig. 2). Participants had to choose which context contained the target as quickly and accurately as possible by clicking on the response alternative using the computer mouse. The stimuli were presented on the screen until the participant gave a correct response (no time limit). If they provided a wrong answer, visual feedback was given on their performance (a red square was shown around the chosen, incorrect alternative) and they were prompted to give a new response until they provided the correct answer. This procedure was put in place to ensure participants would be motivated to actively find the target shape prior to providing an answer, reducing the likelihood of participants randomly guessing to advance through the task. The three embedding contexts were presented at three fixed locations on the screen and their position out of these three locations was picked at random. Stimulus presentation and response registration were controlled using custom software written in C# developed in Visual Studio. The L-EFT was administered individually to participants in small groups during a one-hour session in quiet and slightly darkened computer rooms on Dell Inspiron desktop computers with a 23" monitor. Each group consisted of 15 participants on average. The L-EFT was administered together with two different test batteries during these sessions and
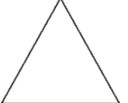

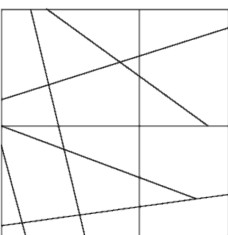 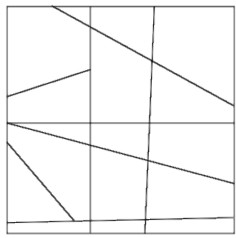 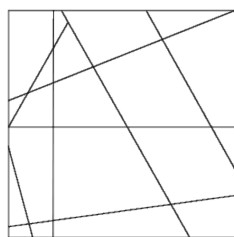

**Figure 2** Matching-to-sample task with three response alternatives.

the order of presentation was counterbalanced with these two test batteries. The L-EFT takes about 10 min to complete.

### Data analysis

For each trial the participant was prompted to provide a new response when their previous response was incorrect, up until the point a correct answer was provided. However, for the data analyses, only the accuracy and response times of the first response on each trial were used. A mixed model logistic regression was used to predict accuracy data on each trial taking into account a chance level of .33. All models were estimated with the R package lme4 (*Bates et al., 2014*; *R Core Team, 2016*). The 99% confidence interval of the odds ratio were estimated as a measure of effect size. An odds ratio equal to 1 indicates no effect. Effects were evaluated against an alpha level of .01.

## Results
### Outliers

On average, participants performed highly accurately on this task ($M = .86$, $SD = 0.08$), taking 2.4 s to correctly detect the target ($SD = 6.54$). None of the participants had more than 15% "fast errors", defined as inaccurate trials for which the respondent answered within 1.5 s. Additionally, no subjects performed below chance level ($<.33$). Therefore, no participants were excluded, and data analyses was performed on the entire sample.

### Speed-accuracy trade-off

There was a moderate speed-accuracy trade-off, $r(253) = .39$, $p < .001$, 95% CI [.28,.49]. Therefore, statistical analyses were performed on accuracy and on response times for accurate trials.

**Table 1  Regression analysis ($N = 255$).**

| Predictors | Accuracy | | | | Log transformed response times | | |
|---|---|---|---|---|---|---|---|
| | Estimate | Z | P | Odds Ratio 99% CI | Estimate | T | P |
| Intercept | 1.64 | 20.45 | <.001 | [4.18, 6.31] | 3.54 | 518.0 | <.001 |
| Target lines | 0.73 | 17.00 | <.001 | [1.86, 2.32] | −0.06 | −24.4 | <.001 |
| Continued lines | −1.70 | −25.43 | <.001 | [0.15, 0.22] | 0.22 | 52.4 | <.001 |
| Closure[a] | −0.10 | −1.47 | .14 | [0.77, 1.07] | −0.04 | −10.0 | <.001 |
| Symmetry[b] | 0.35 | 5.40 | <.001 | [1.20, 1.68] | −0.03 | −6.0 | <.001 |
| Target lines × Continued lines | 0.33 | 8.76 | <.001 | [1.27, 1.54] | −0.05 | −17.0 | <.001 |
| Closure × Continued lines | 0.18 | 2.86 | .00 | [1.02, 1.40] | −0.02 | −4.7 | <.001 |
| Symmetry × Continued lines | −0.14 | −2.26 | .02 | [0.74, 1.02] | −0.01 | −1.9 | .06 |

**Notes.**

Number of target lines and number of continued lines were standardized. Due to this procedure the odds ratio can be interpreted as a measure of effect size independent of measurement scale.

[a]Closed shapes = 1, open shapes = 0.

[b]Symmetry = 1, asymmetry = 0.

### *Effect of stimulus features*

The logistic regression model for accuracy and the linear regression model for response times that were tested, included the fixed effects of the number of target lines, closure, symmetry and the number of continued lines. Additionally, the pairwise interactions between the number of continued lines and the other predictors were included in the model. Furthermore, a random intercept for participants was included. For response times, the data were logarithmically transformed to correct for a positive skew. The results of the logistic and linear regression are reported in Table 1.

For accuracy, there was a significant effect of number of target lines, number of continued lines, and symmetry. Accuracy was higher for symmetric shapes, shapes consisting of more lines, and for contexts with fewer continued lines. There was also an interaction between the number of continued lines and target lines, and an interaction between the number of continued lines and closure. The significant interaction between continued lines and target lines indicates that the effect of continued lines was weaker for target shapes that were made up of more lines. The main effect of closure and the interaction between continued lines and symmetry were not significant.

The analysis on response times mostly mirrored the accuracy results: a stimulus feature that increased accuracy also resulted in faster response times for target detection. However, for the effect of closure the results of response times were inconsistent with the observed trend for accuracy: response times were significantly faster for closed shapes in comparison to open shapes, while accuracy was slightly lower for closed shapes than open shapes. The interaction of number of target lines and the number of continued lines for accuracy and median response times on accurate trials are visualized in Fig. 3.

### Discussion

The L-EFT figures were designed to test how particular features of a target shape (closure, symmetry and complexity) and embedding context (the number of continued lines) influence task difficulty. The results supported the notion that symmetry made a target

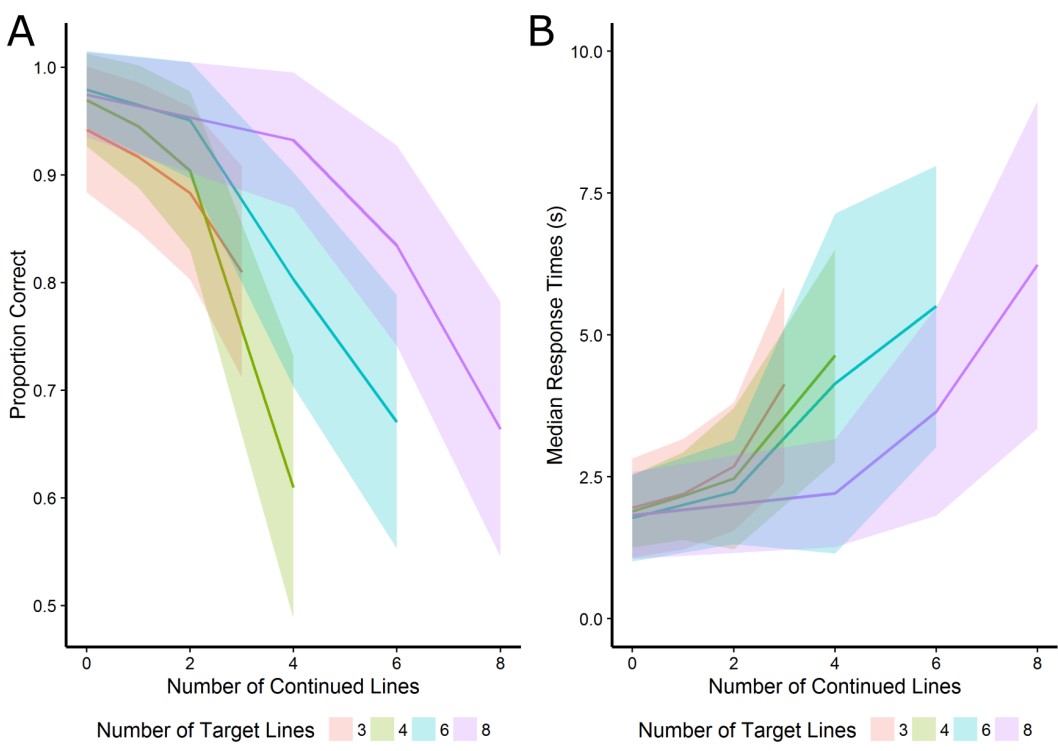

**Figure 3** **The interaction of number of target lines and the number of continued lines for accuracy and median response times on accurate trials.** The shaded area represents half a standard deviation.

easier to find (or more difficult to embed). There was also a very clear influence of the number of continued lines, such that targets with more lines continued into the embedding context were harder to find. This clearly supports the idea that the good continuation between the target and its context is an important factor in influencing the strength of the perceptual embedding. The effect of target complexity—operationalized as an increasing number of target lines—did not affect performance in the expected direction. That is, shapes with more target lines were easier to detect. This could be because a shape with more target lines is a more unique occurrence in the embedding context, therefore enhancing the likelihood of target detection.

The results also indicated that the influence of good continuation interacted with some of the stimulus features of the figure. The interaction between the number of continued lines and the number of target lines shows that accuracy mostly depended on the proportion of target lines that are continued into the context (see Fig. 3). Additionally, the interaction of the number of continued lines and closure was significant. One could argue that the "good Gestalt" of a closed shape rendered it less susceptible to the influence of continued lines, but this interpretation is hard to reconcile with the fact that symmetric shapes (which should also form a good Gestalt) did not render these shapes less susceptible to the manipulation of continued lines and with the absence of a main effect of closed target shapes (which suggests that closed shapes were no better Gestalt compared to open shapes within this L-EFT stimulus set).

## EXPERIMENT 2

Experiment 2 aimed to replicate the results of Experiment 1 and to optimize certain task features of the L-EFT. More specifically, a time limit was introduced for each trial to increase the task difficulty and reduce the speed-accuracy trade-off.

### Materials & methods
#### *Participants*
A sample of 188 undergraduate psychology students participated in this study for course credits. All participants were naïve to the purpose of the study. None of the participants participated in Experiment 1. The median age of subjects was 18 years ($SD = 1.99$). The sample was primarily female (79%). All procedures performed in this study were in accordance with the ethical standards of the institutional ethical committee and approved by the ethical committee of the KU Leuven university (SMEC approval code: S58409) as well as in accordance with the 1964 Helsinki declaration and its later amendments or comparable ethical standards. Written informed consent for each participant was obtained prior to testing.

#### *Stimuli*
The same target and context shapes as in Experiment 1 were used.

#### *Procedure*
The task procedure of the L-EFT was slightly altered compared to the L-EFT version used in Experiment 1. All 64 figures were presented twice in a randomized order (128 trials). Subjects were asked to provide one answer and were not given the opportunity to change their initial response. No response feedback was provided. A forced choice matching-to-sample paradigm was used, similar to Experiment 1, but the target and the three response alternatives were only presented for a limited duration (3 s). The stimulus presentation and response registration were controlled using custom software written in Python using PsychoPy (*Peirce, 2007*).

#### *Data analysis*
To evaluate whether the results of Experiment 1 could be replicated, the same analyses as in Experiment 1 were performed. In line with the changes made to the task procedure, the accuracy and response times of the first and only response of the participant for each trial was taken into account.

### Results
#### *Outliers*
On average, participants were highly accurate on this task ($M = .82$, $SD = 0.06$), taking 2.01 s to detect the target ($SD = 0.64$). Participants with a high number of fast errors on the task were deemed unmotivated and were removed from the data (1.59% of participants). Fast errors were defined as inaccurate trials in which the respondent answered within 1.5 s. A cut-off of 15% fast errors was used. None of the subjects performed below chance level ($<.33$).

**Table 2  Regression analysis ($N = 185$).**

| Predictors | Accuracy | | | | Log transformed response times | | |
|---|---|---|---|---|---|---|---|
| | Estimate | Z | P | Odds Ratio 99% CI | Estimate | T | P |
| Intercept | 1.28 | 22.13 | <.001 | [3.10, 4.17] | 0.28 | 36.75 | <.001 |
| Target lines | 0.82 | 23.88 | <.001 | [2.08, 2.48] | −0.06 | −46.82 | <.001 |
| Continued lines | −2.24 | −36.67 | <.001 | [0.09, 0.13] | 0.16 | 68.69 | <.001 |
| Closure[a] | −0.25 | −4.80 | <.001 | [0.68, 0.89] | −0.04 | −16.93 | <.001 |
| Symmetry[b] | 0.40 | 7.77 | <.001 | [1.31, 1.70] | 0.00 | 2.09 | .04 |
| Target lines × Continued lines | 0.39 | 12.00 | <.001 | [1.35, 1.60] | −0.03 | −21.49 | <.001 |
| Closure × Continued lines | 0.41 | 7.65 | <.001 | [1.31, 1.73] | −0.00 | −0.41 | .68 |
| Symmetry × Continued lines | 0.12 | 2.31 | .02 | [0.99, 1.29] | −0.00 | −1.00 | .32 |

**Notes.**

Number of target lines and number of continued lines were standardized. Due to this procedure the odds ratio can be interpreted as a measure of effect size independent of measurement scale.

[a]Closed shapes = 1, open shapes = 0.

[b]Symmetry = 1, asymmetry = 0.

### Speed-accuracy trade-off

There was no significant correlation between mean accuracy and median response times of participants, $r(183) = .01$, $p = .85$, 95% CI $[−.13, .16]$.

### Effect of stimulus features

The same models were used as in Experiment 1. The results of the logistic and linear regression are reported in Table 2. Response times were logarithmically transformed to correct for a positive skew. The results of the accuracy analysis were highly consistent with the findings of Experiment 1. Detection is better for symmetric shapes with more target lines and for shapes with fewer lines continued into the embedding context. In contrast to the results of Experiment 1, the effect of closure on accuracy now proved significant (even though the manipulation of closure was the same as in Experiment 1). Accuracy was higher for open than closed shapes. The two-way interaction between the number of target lines and continued lines was significant, which indicates that the effect of continued lines was weaker for shapes with more target lines. The interaction between closure and continued lines was also significant and indicates that the effect of continued lines is weaker for closed than for open target shapes. The interaction between continued lines and symmetry was not significant.

The results of the response times are mostly consistent with the accuracy data. However, the main effect of closure on response times indicated that participants were slightly faster to detect closed shapes compared to open shapes, which contradicts the accuracy results. Additionally, there was no significant effect of symmetry on response times and there was no significant interaction between the number of continued lines and the target features of closure and symmetry. The interaction of the number of target lines and number of continued lines for accuracy and median response times on accurate trials is visualized in Fig. 4.

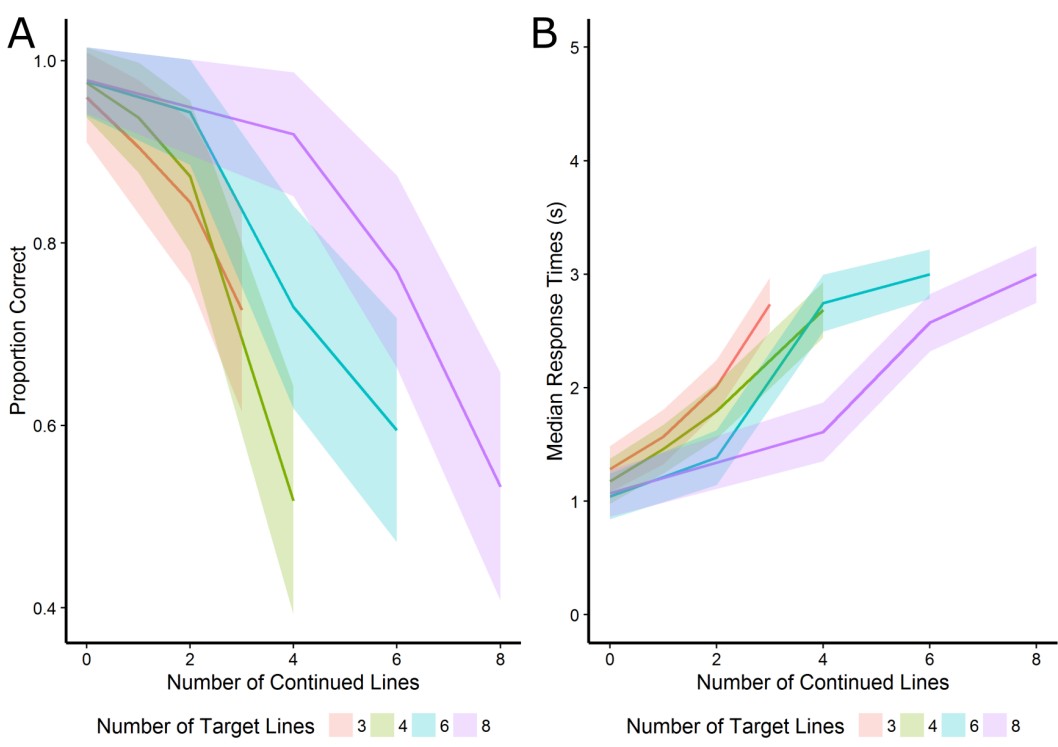

**Figure 4** **The interaction of the number of target lines and number of continued lines for accuracy and median response times on accurate trials.** The shaded area represents half a standard deviation.

## Discussion

Experiment 2 aimed to replicate the results of Experiment 1 and optimize certain task features of the L-EFT. A time limit was introduced for each trial to increase the difficulty and reduce the speed-accuracy trade-off. While the time limit did not affect the task difficulty, it did reduce the speed-accuracy trade-off. The results of Experiment 2 are in line with those of Experiment 1. Again, there was a clear influence of symmetry and the number of lines of the target shape that were continued as lines in the embedding context. The effect of the number of target lines was similar to Experiment 1, although this effect contradicted our initial expectations. Again, there was an interaction between closure and the number of continued lines, and the number of lines in the target figure and the number of those lines that are continued. As in Experiment 1, no interaction was found between the number of lines continued and the symmetry of the figure. The only inconsistent finding between both experiments was the main effect of closure on accuracy, for which the accuracy and response time effects also seem to point in different directions.

## CONCLUSIONS

Since its development as a tool to measure individual differences in cognitive style, the EFT has become a popular test to measure local and global perceptual style, both with regard to typical development and different clinical populations (*Cribb et al., 2016*; *Panton, Badcock & Badcock, 2016*). However, *Milne & Szczerbinski (2009)* have shown that the

factor underlying EFT performance is better represented by a more narrow construct of *disembedding* instead of a local/global perceptual style. The interpretation of good EFT performance as a reflection of a local perceptual style is problematic, because it is unclear how a local perceptual style would affect the level of perceptual grouping an individual tends to perform and it is also unclear what the local and global levels are in the embedded figures. Our new version of the EFT does not necessitate the interpretation of the results as either a reflection of a local/global perceptual style, but assesses how people are affected by different target and context features. With that purpose, the L-EFT was developed with a focus on a controlled and parameterized stimulus set, made freely available to others that allows to test to what extent embeddedness of target shapes depends on different features of the target and context.

Three key factors relating to the target shape were explicitly manipulated in two experiments, namely the complexity of the shape (defined simply as the number of lines making up the target), its symmetry (specifically around the vertical axis) and whether or not the target shape was an open or closed shape. Across the two experiments, there was consistent evidence that symmetric shapes were easier to detect. This suggests that the factors that contribute towards the formation of a *good Gestalt* can also influence embeddedness of shapes. Unexpectedly however, evidence was mixed with regard to closure as the main effect of closure was not significant in Experiment 1 and had inconsistent effects for accuracy and response times in Experiment 2.

Whilst previous evidence has already indicated that closed shapes do form a good Gestalt (*Elder & Zucker, 1993*; *Kovács & Julesz, 1993*), the openness of the target shape did not influence target shape embedding in the L-EFT as predicted. It is not entirely clear why symmetry influenced resistance to embedding, while closure did not. It could be that symmetry is a more independent feature of the configuration of line segments compared to the closure of a target shape in the embedded figures test. That is, in contrast to symmetry, closure is inherently ill-defined in embedding contexts, because one does not yet know which line segments belong to the figure and which line segments belong to the context. Therefore closure may not be able to act as a protective factor against embedding. The effect of target complexity was also not in line with our expectations. The target shapes made up of more lines were easier to detect, which may be explained by the fact that the observer has more information about the target shape and can therefore find the shape more easily.

However, the latter result does not imply that the complexity of the target shape did not affect embeddedness. That is, target complexity depends on more than just the number of lines a shape consists of. Asymmetric and irregular shapes are more complex than symmetric regular shapes, therefore the complexity of the target contour in our stimulus set did not correlate perfectly with the number of target lines. If we define target complexity more broadly than just the number of target lines, the results could be interpreted as showing that target complexity indeed contributed to a higher degree of embeddedness, because symmetric shapes were more easily detected. Therefore, we can conclude that target shapes that form better Gestalts are more easily detected in the embedded figures task. Whether this effect arises from a better representation of the Gestalt in working memory (whilst

searching for the target shape) or whether this effect relates to a better representation of the target shape in the embedding context itself cannot be disentangled based on the current study. It seems likely that both aspects contribute to better target detection.

In addition to manipulating the nature of the target shape, we also systematically manipulated the level of good continuation between the target and context shape by varying the number of lines of the target figure that were continued into the embedding context. Across the two experiments, we found clear evidence that this manipulation was highly effective in influencing the strength of embedding, such that more continued lines led to both lower accuracies and longer response times. That is, across both experiments we found that detection of the target became approximately five to ten times more likely when the number of continued lines was reduced by 1 standard deviation. This finding supports the notion that good continuation is one of the most important factors in influencing the effectiveness of perceptual embedding.

The specific means of manipulating good continuation in terms of the number of continued lines was motivated by *Rao & Ballard*'s (*1999*) predictive coding account of 'end-stopping.' End-stopping is a property of certain neurons in the primary visual cortex that fire in response to an edge that 'ends' at a particular point in space, but stop firing when that edge is continued into a longer line. Rao and Ballard argued that this response property (end-stopping) actually indicates that neurons which behave in this way are not firing to the presence of an edge, but are firing to signal an 'error' based on a prediction at higher areas that that edge should be part of a longer line. Our results regarding the effect of good continuation could be interpreted as suggesting that the detectability of individual lines (and the capacity for those lines to be grouped into a target shape) is reduced when they can be interpreted as longer lines, and the cells in early visual areas no longer signal the 'error' of these 'unpredicted' shorter lines. This is of course only one of many possible explanations for this effect, however it seems pertinent to outline this potential explanation here as the EFT test has been consistently related to ASD (*Cribb et al., 2016*), and some theories of ASD conceptualize perceptual differences in terms of an overweighing of prediction errors (*Van de Cruys et al., 2014*). Clearly, further research would be needed to test the validity of this interpretation and its implications for ASD.

Our results clearly show how and to what extent perceptual grouping is involved in the effective embedding of different line shapes on a group level. The fact that different perceptual properties influence the strength of embedding suggests that the traditional EFT is likely to offer a complex aggregated measure of numerous different perceptual (in addition to cognitive) processes. That is, inter-individual differences in EFT performance may relate to differences in the effect of target and context features. For instance, the manipulation of target features could have a stronger effect on people who have a more limited memory span, because targets that form a good Gestalt could benefit people with small memory spans more. Additionally, inter-individual differences in sensitivity to good continuation may also be a source of EFT variance given the fact that good continuation had a strong effect on target embeddedness on a group level. In addition to these potential sources of inter-individual differences, differences may also arise from more pure cognitive processes, such as differences in fluid intelligence or cognitive flexibility. In future work

we will test whether inter-individual differences in this complex aggregate score primarily reflect 'perceptual style' (as often assumed) or other perceptual or cognitive constructs by testing the extent to which performance on the traditional EFT and our new L-EFT task correlates with inter-individual differences on different perceptual, executive and problem solving tasks.

### Funding

Lee de-Wit and Rebecca Chamberlain received a post-doctoral fellowship from the Research Foundation–Flanders (FWO). This work was also supported by the Methusalem program by the Flemish Government (METH/08/02, METH/14/02), awarded to Johan Wagemans. The funders had no role in study design, data collection and analysis, decision to publish, or preparation of the manuscript.

### Grant Disclosures

The following grant information was disclosed by the authors:
Research Foundation–Flanders (FWO).
Methusalem program: METH/08/02, METH/14/02.

### Competing Interests

The authors declare there are no competing interests.

### Author Contributions

- Lee de-Wit conceived and designed the experiments, wrote the paper.
- Hanne Huygelier conceived and designed the experiments, performed the experiments, analyzed the data, wrote the paper, prepared figures and/or tables.
- Ruth Van der Hallen and Rebecca Chamberlain conceived and designed the experiments, performed the experiments, reviewed drafts of the paper.
- Johan Wagemans reviewed drafts of the paper.

### Human Ethics

The following information was supplied relating to ethical approvals (i.e., approving body and any reference numbers):

The Social and Societal Ethics Committee of the KU Leuven (SMEC) approved this study (Ethical Application Reference: S58409).

### Data Availability

The raw data has been supplied as a Supplementary File.

### Supplemental Information

Supplemental information for this article can be found online at http://dx.doi.org/10.7717/peerj.2862#supplemental-information.

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
