# Peer review of "Developing the Leuven Embedded Figures Test (L-EFT): testing the stimulus features that influence embedding"

_PeerJ, doi:10.7717/peerj.2862_

## Round 0.1 · original submission · Minor Revisions

I have received reviews from two experts in the field. Both were quite positive about your manuscript.

Reviewer 1 requested that you provide a more substantive overview of the theoretical context into which your piece fits, with a special focus on how the individual differences observed in the EFT map onto performance in your task. I agree that this would be quite helpful to the reader. Reviewer 1 also suggests carrying out a factor analysis on your results in order to assess the relationship between the L-EFT and the EFT and set the stage for future explorations into the mechanisms underlying embedding. While I will not insist upon this specific analysis, I do encourage you to try to address individual differences in performance on your task.

This focus on individual differences is echoed in the concerns of Reviewer 2 (James Pomerantz), who notes that your sample was dominated by females. Perhaps this imbalance could motivate some exploration of individual differences within your sample.

The remainder of the reviews speak for themselves. Your manuscript is exceptionally well-written, and I see no reason that it shouldn't be accepted for publication following some minor revisions. Bravo.

Reviewer 1 ·

Basic reporting

The article appears to adhere to the PeerJ policies and to be appropriate in terms of basic reporting.

Experimental design

The experimental methods are appropriate for this type of research.

Validity of the findings

The findings appear to be valid. Suggestions regarding further statistical analysis and also qualification of interpretations of the findings are included in the General Comments.

Additional comments

The study concerns the Embedded Figures Test (EFT), which was invented to study individual differences in “field in/dependence” and more recently to study such differences in autistic individuals with a possibly related, but different, type of interpretation—local/global perceptual style. The authors appear to have two primary goals: to systematically manipulate aspects of Embedded Figures stimuli and also to do so with respect to a known visual cortical mechanism (end-stopping). The study is worthwhile and with respectable intentions. Nevertheless, some improvements in analyses and interpretations are recommended prior to publication in PeerJ (see following comment sections). Once these comments are addressed, I feel the article is worthy of publication.

Analysis-related:
Given that one of the main goals of the L-EFT is to identify inter-individual differences in performance, it might make sense to perform a factor analysis on your data. This could serve to confirm that the L-EFT is assessing the same dimensions as the EFT, or to explore the underlying factors assessed by a more rigorously parameterized perceptual test. While identifying the mechanisms underlying differences in performance is left to future studies, this first step may motivate additional research.
A clarification on lines 149-150, it sounds as though in Experiment 1 you are only performing statistical analysis on accuracy for accurate trials, but I presume you meant you are only analyzing RT for accurate trials.

Interpretation-related:
First, the authors report that (vertical mirror) symmetry conferred an advantage in the task such that symmetric figures were easier to detect. In contrast, there was not much (if any) effect of “complexity” in terms of the number of elements comprising a shape. While the latter finding is not in question, I encourage the authors to interpret their symmetry effect in terms of complexity, because when the number of “parts” of a configuration is held constant, complexity still varies with symmetry such that an increasing number of symmetries (not just mirror, but also rotational symmetry) results in decreased complexity. What I’m trying to say is that the discussion of complexity should include not only the null effect of number of line segments (which had not appreciable effect) but also the effect of symmetry, which implies that increasing complexity (by decreasing symmetry) does in fact have an effect such that more complex shapes are less detectable. Obviously their operationally defined manipulation of complexity is fine, but the symmetry findings warrants a clarification of the potential effects of complexity manipulated with respect to symmetry or possibly other means (e.g. complexity of the entire display). Similarly, the complexity of a shape is also related to the complexity of its bounding contour, such that less “smooth” contours are more complex—and inverse relationship between complexity and good continuation. In short, while the authors reasonably manipulated complexity, it is worth discussing the relationship between complexity, symmetry and good continuation since the latter two had notable effects on performance of the task.


Second, the authors begin with a clarification about the original intended utility of the EFT (in relation to cognitive style) and also comment on how in more recent work using the EFT with autistic individuals the emphasis has been on local-global processing. It would be good to elaborate a bit further on why this might be valid or invalid, and more importantly, to be clearer in the Discussion section about how the current results relate to this. My reading is that the authors are suggesting a better understanding of which stimulus properties influence EFT performance, and how such stimulus property manipulation might relate to neural mechanisms (e.g. end-stopping), but it is less clear how this would relate to individual differences in performance—are the authors suggesting that individual differences in “perceptual style” can be traced to basic visual cortical mechanisms? Why the focus on end-stopping and not more on top-down influences, which I realize is not the focus of the study but plays a critical role (according to some, e.g. M. Peterson and colleagues)… I’d like more discussion of this so that readers, especially those who are not experts in visual perception, have a richer context within which to digest the results.


Finally, given neuroimaging studies of the task (e.g. by Ring et al., 1999, as an early example), what is the relationship between EFT performance, the manipulations conducted in the current study, and visual mechanisms in ventral visual cortex? That is, in addition to the prospective contribution of low-level visual mechanisms to EFT performance, what higher-level visual mechanisms (in addition to more “cognitive” factors) could be important to the task?

·

Basic reporting

Excellent. Very clearly written, appropriate lit review. The figures are almost too light to see well, so I would recommend bolding the lines, so long as that would not distort the accurate reporting of what the stimuli were. A quick illustration showing the number of continued lines, etc. might be helpful. The Materials & Methods section of Experiment 2 might be shortened by including only the difference from Experiment 1.

Experimental design

Quite solid. The second experiment provides a replication of the first, which is especially important in these times when effects so often fail to replicate. The only note I would make on the experimental design was the high proportion of females; normally I wouldn’t see this as a possible issue, but given that the focus of the EFT has traditionally been on individual differences, this could matter here.

Validity of the findings

Again I give this manuscript high marks here. The logic and conclusions are quite clearly stated and justified, and needed future work is outlined. I see this experiment as breaking new ground, so I look forward to future work from these investigators.

Additional comments

The authors are quite correct, in my opinion, when they say that few researchers have tried to determine what stimulus factors lead to hidden or embedded figures. This is one reason why many included myself have not held the EFT in high regard. This study corrects that, by going after the likely suspects and putting them to fair, empirical tests. Those likely suspects include closure, proximity, similarity, complexity, and good continuation.

One question I have is how perceptual the effects found here may be. Consider trying to find a target equilateral triangle embedded in a background. If that symmetric target is easier to detect than an asymmetric target, is it because the symmetric one is easier to perceive in the background or because it is easier to remember while searching? (Note that while searching the background for a target, the target being searched for must be held in memory.) Gestalt effects are nearly as robust in memory as in perception – both may result from shorter codes used for better Gestalts - so for purposes of the L-EFT it really isn’t critical where the effect is arising in the processing system.

It’s possible that the mixed effects of closure (line 288) results from all the displays containing lots of closure, making it difficult to find the needle in a haystack, so to speak.

---

## Round 0.2 · accepted · Accept

I have reviewed the changes to your manuscript and believe that you have addressed all of the reviewers (admittedly minimal) critiques. Your paper will make an important contribution to the literature, and I anticipate that it will spur much future research. Bravo!